# A bionic self-driven retinomorphic eye with ionogel photosynaptic retina

Xu Luo[1,4], Chen Chen[1,4], Zixi He[1,4], Min Wang[1], Keyuan Pan[1], Xuemei Dong[1], Zifan Li[1], Bin Liu[1], Zicheng Zhang[1], Yueyue Wu[1], Chaoyi Ban[1], Rong Chen[1], Dengfeng Zhang[1], Kaili Wang[1], Qiye Wang[1], Junyue Li[1], Gang Lu ⬤[1], Juqing Liu ⬤[1] ✉, Zhengdong Liu[1] ✉ & Wei Huang[1,2,3] ✉

Bioinspired bionic eyes should be self-driving, repairable and conformal to arbitrary geometries. Such eye would enable wide-field detection and efficient visual signal processing without requiring external energy, along with retinal transplantation by replacing dysfunctional photoreceptors with healthy ones for vision restoration. A variety of artificial eyes have been constructed with hemispherical silicon, perovskite and heterostructure photoreceptors, but creating zero-powered retinomorphic system with transplantable conformal features remains elusive. By combining neuromorphic principle with retinal and ionoelastomer engineering, we demonstrate a self-driven hemispherical retinomorphic eye with elastomeric retina made of ionogel heterojunction as photoreceptors. The receptor driven by photothermoelectric effect shows photoperception with broadband light detection (365 to 970 nm), wide field-of-view (180°) and photosynaptic (paired-pulse facilitation index, 153%) behaviors for biosimilar visual learning. The retinal photoreceptors are transplantable and conformal to any complex surface, enabling visual restoration for dynamic optical imaging and motion tracking.

After evolving for millions of years, most bio-eyes possess photo-receptor arrays in a curved form with broadband photoperception, neuromorphic vision and retinal transplantation functions, which enable the eyes to efficiently capture and process visual signals with ultralow power consumption. Such remarkable features inspire the emulation of artificial eye with hemispherical structures and retinal photoreceptors made of materials such as silicon, perovskite nanowires, and other two-dimensional heterostructures[1–3]. The human-made eyes present wide field-of-view (FOV) and high resolution imaging with power supply, they normally require integration with additional computing hardware for signal processing[4,5]. This near-sensor computing architecture containing separate sensing and processing units inevitably produce inefficient speed, redundant data, and high energy consumption. Recently, artificial eyes based on self-powered photosynapses have been constructed, the photoreceptors exhibit a limited light response range[6,7]. Additionally, their rigid components make them complex to manufacture and insufficiently flexible in conformation, resulting in high production costs and limited application scenarios. To address these challenges, it is necessary to develop advanced biomimetic eyes with in-sensor computing capability for machine vision, particularly self-driving, retinal transplantation and high conformal capabilities[8–11].

Here we demonstrate an artificial self-powered hemispherical retinomorphic eye (SHR-E) consisting of heterobilayer ionogel pillarforest as retinal photoreceptors, and experimentally investigate neuromorphic photoperception, retinal transplantation and visual

[1]Key Laboratory of Flexible Electronics (KLoFE) & Institute of Advanced Materials (IAM), School of Flexible Electronics (Future Technologies), Nanjing Tech University (NanjingTech), Nanjing, China. [2]Frontiers Science Center for Flexible Electronics, Institute of Flexible Electronics (IFE), Northwestern Polytechnical University, Xi'an, China. [3]State Key Laboratory of Organic Electronics and Information Displays, Nanjing University of Posts and Telecommunications, Nanjing, China. [4]These authors contributed equally: Xu Luo, Chen Chen, Zixi He. ✉e-mail: iamjqliu@njtech.edu.cn; iamzdliu@njtech.edu.cn; iamwhuang@njtech.edu.cn

restoration for real-time optical imaging and motion tracking. The schematic of the octopus eye-inspired SHR-E system is displayed in Fig. 1a. Beyond structural similarity in lens, iris, optic nerve, retina pigment and photoreceptors within human eye, octopus eye has a verted retina in which the photoreceptor layer lies in front of the optic nerve, leading to no blind spots[12]. In our SHR-E design, the photosensitive heterojunction is developed through selective polypyrrole[13] nanoparticles (PPy-NPs) doping of ionogel pillar, a pillar array can be directly implanted onto the surface of hyaline ionogel hemisphere toward incident light to form retina-like pillar-forest, with the ability of optical-to-electrical converter and neuro-electric plasticity simultaneously (Fig. 1b). The self-powering trait is mainly provided by photothermoelectric induced ion drift within the ionogel, which is different from the previous self-powered optical synapses induced by photovoltaic effects (Supplementary Table 1), making it hugely potential for energy-saving autonomous sensing technology[14]. Moreover, all-soft components endow the retina with excellent conformal and stretchable capability that can adhere to any objects with complex geometries (Fig. 1d, see below). Our approach offers a facile and effective route to construct zero-powered photodetectors, specially transplantable organ-like conformal retinal photoreceptors.

## Results

### Fabrication process and photosensitive characteristics

The chemical synthesis of each compound and junction fabrication process used to experimentally create SHR-E is detailed in Methods section. Like their biological counterparts, the synthetic pure iono-gel is highly optical transparency (93%) and soft elasticity, lithium cations (Li+) and bis(trifluoromethanesulfonyl)imide anions (TFSI-) fixed in the ionogel are mobile for ionic current (Supplementary

Figs. 1 and 2). The ionic movement as charge carriers has been uti-lized in emerging diodes and transistors[15,16], making them as an ideal nerve fiber to transmit neuroelectric signals. By selectively doping the photosensitive PPy-NPs as pigment in ionogel, the color changes from transparency to black, enabling their spectra to overlap for ultra-broadband absorption in the ultraviolet-visible-near infrared (UV-vis-NIR) region (Supplementary Fig. 3). Under light exposure, ions migrate from PPy-gel to pure-gel within heterojunction, resulting in an imbalanced ion concentration due to different migration rates between cations and anions. With the synergistic assistance of photothermal and thermoelectric conversion pro-cesses within each component[16–18], a temperature gradient can pro-duce ionic current (Fig. 1c). Because of the mobility difference of Li+ and TFSI- species[19], a larger number of Li+ ions infiltrate into the pure-gel, leading to an excess of positive ions over negative ions. As a result, there are more TFSI- ions than Li+ ions present in PPy-gel, resulting in a built-in electric field in heterojunction, with the field direction pointing from pure-gel toward PPy-gel. Meanwhile, the field also impedes Li+ ions diffusion while accelerating TFSI- ions diffusion until reaching equilibrium state. The built-in electric field entrusts the ionogel heterojunction with photoreception ability self-driven by the photothermoelectric effect.

To examine the self-powered photodetecting prototype of iono-gel heterojunction, we design a photoelectric response experiment in which photocurrent and response time (RT) are measured at the het-erointerface portion. Photoresponse phenomenon is analogous to the effect of light shining into retina. A time-resolved photogenerated current is illustrated in Fig. 2a under a single light pulse (7.2 μW mm$^{-2}$, 365 nm) without external bias loading, with an 11.18 s rise time ($\tau_r$) and 64.36 s fall time ($\tau_f$). Such delayed photoconductivity is likely attrib-uted to the slow kinetics of ion flux compared with electric

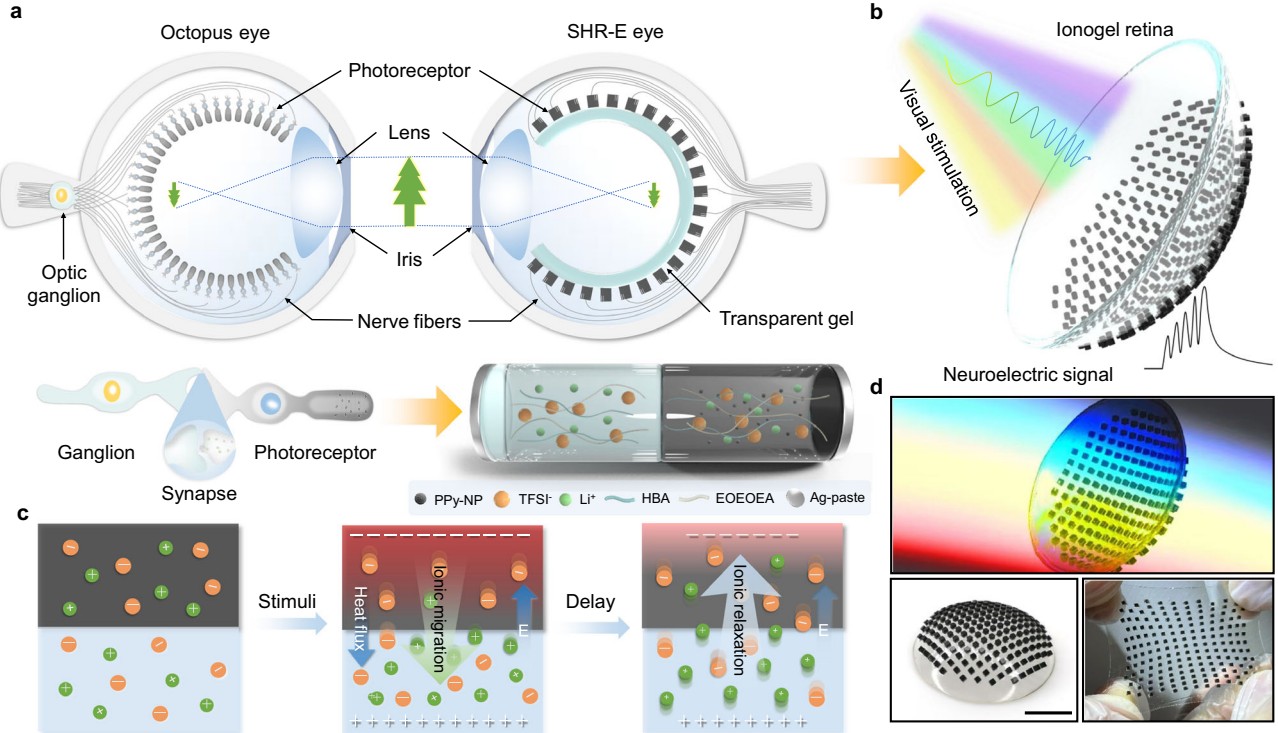

**Fig. 1 | Bioinspired SHR-E structure with ionogel heterojunction retina.**
**a** Anatomy of an octopus eye and biomimetic SHR-E eye with ionogel photo-receptor (SHR-E: self-powered hemispherical retinomorphic eye, PPy-NPs: poly-pyrrole nanoparticles, TFSI: bis(trifluoromethanesulfonyl)imide anions, HBA, 4-Hydroxybutyl acrylate, EOEOEA: 2-(2-Ethoxyethoxy) ethyl acrylate). **b** Schematics of our hemispherical retina with neuromorphic signal processing from ionic heterogel pillarforest. **c** Process mechanism of optical-to-electrical converter (E: electric potential). **d** Photographs of the hemispherical retina with multicolor light exposure, high surface conformal and stretchability. Scale bar, 1 cm.

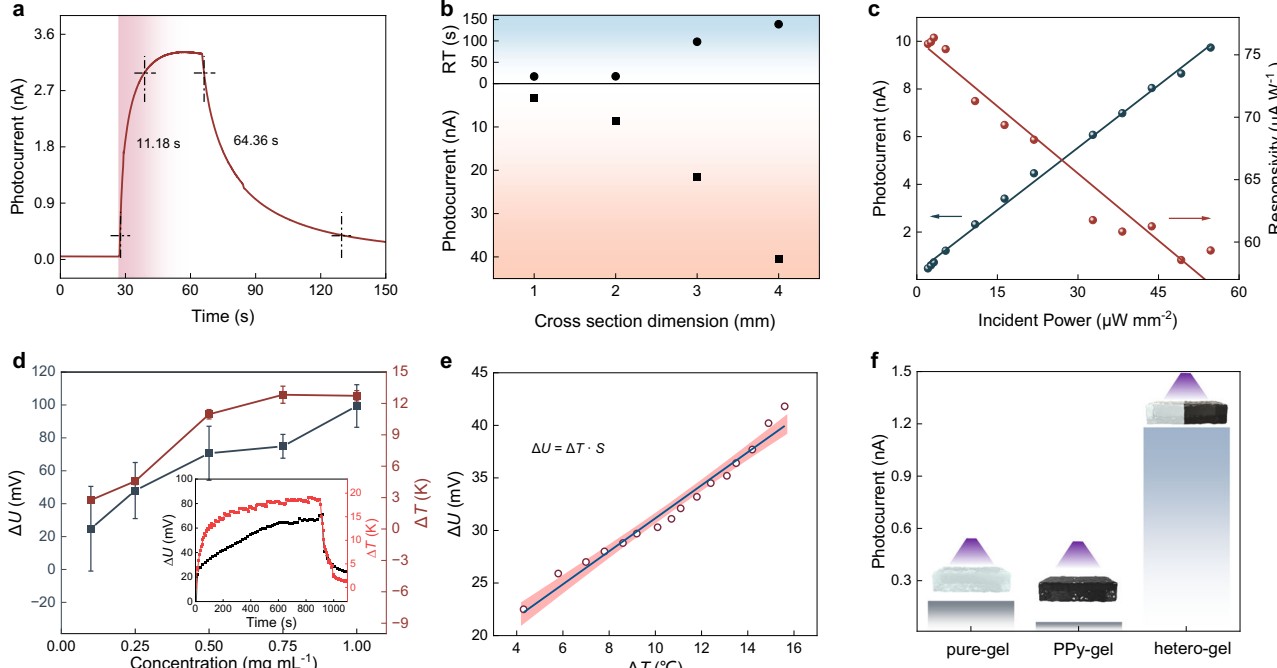

**Fig. 2 | Self-driven photodetection and photothermoelectric characterization of ionogel heterojunctions. a** Photoresponse of the heterojunction exposed to 365 nm light without an external power supply. **b** Size effect on photocurrent and responsive time (RT: responsive time). **c** Illumination intensity-dependent photocurrent and responsivity. **d** Photothermoelectric parameters versus doping concentration of an individual heterojunction, the inset illustrates the voltage and temperature change over expose time ($\Delta U$: voltage difference, $\Delta T$: temperature difference). Error bars represent standard deviations. **e** Voltage difference-temperature difference curves for Soret effect mechanism ($S$: Soret coefficient). **f** Comparison of photocurrent of the heterojunction, pure and doped ionogels under light illumination.

migration[20,21], making the heterojunction a promising synaptic photoreceptor for retinomorphic emulation. Both the excitatory postsynaptic current (EPSC) and $\tau_f$ depend on the cross-sectional area of junction pillar (Fig. 2b), with a smaller area producing lower current and faster decay than the larger one, meaning a quicker response with weaker neuroelectric signal in the finer pillar-like photoreceptor. Thinning the pillar can also enhance array density and, in turn, enable high resolution imaging. Noted that this bionic photoreceptor is similar to biological one, as photocurrent and responsivity can be modulated by light intensity (Fig. 2c), with positively and inversely linear dependency of incident light power, respectively. Impressively, the junction exhibits a high responsivity of 75.87 μA W$^{-1}$ under an excitation of 2.08 μW mm$^{-2}$, ensuring its capability of weak light detection.

Besides photoelectric response inspection, it is essential to measure photothermal and thermoelectric effect to explore self-powered operation. As presented in Fig. 2d, a temperature difference ($\Delta T$) between the doped and undoped portions is built up after light exposure (Supplementary Fig. 4), which drives directional ion migration toward temperature gradient drop, and subsequently generates an imbalanced ion distribution induced by the difference of Li$^+$/TFSI$^-$ migration speed, eventually leading to an electric potential or voltage difference ($\Delta U$) (Supplementary Fig. 5). The $\Delta T$ and $\Delta U$ value both monotonically increase with increasing dopant concentration, because the similar evolution trends are observed in the time-dependent $\Delta T$ and corresponding $\Delta U$ characters. This behavior is consistent with the Soret effect[22,23], the voltage gradient is linearly proportional to the temperature gradient, with a relationship described as $\Delta U = S \cdot \Delta T$, where $S$ is the Soret coefficient, which normally determines the magnitude of thermodiffusion in ionogel, with a value of 1.57 mV K$^{-1}$ at 1.0 mg mL$^{-1}$ doping concentration (Fig. 2e). An intercept voltage presents in initial stage after illumination, which is probably ascribed to hot electron emission from the silver paste

electrode (Supplementary Fig. 6)[24,25]. Moreover, a significant enhancement of photocurrent in heterostructure was observed compared to the two ionogels with or without dopants (Fig. 2f and Supplementary Fig. 7), certifying that the implantation of photosensitive PPy-NPs plays a critical role in efficient photo perception. Furthermore, temperature-dependent photocurrent behaviors are observed in our heterojunction (Supplementary Fig. 8). The EPSC is more sensitive to high temperature due to the enhanced activity and accelerated migration rate of ions within the gel at elevated temperature.

## Photonic synaptic characteristics

Based on the above photoelectric analysis, the photoreceptor has ultra-broadband optical absorption, highly sensitive photodetection with persistent photoconductivity, and self-driven operation, endowing the junction to encode multiwavelength optical signals with photosynaptic plasticity (Supplementary Fig. 9). Figure 3a presents the EPSC of the photoreceptor, these signals enhance steadily with the increase of light pulse number, inducing short-term presynaptic enhancement with evident paired-pulse facilitation (PPF) behavior in the overall UV-vis-NIR range. For a bionic photosynapse, both of spike-intensity-dependent plasticity (SIDP) and spike-number dependent plasticity (SNDP) are observed in Fig. 3b, c. By raising intensity and number of spikes, all refresh EPSC signal levels of the latter spike are stronger than that of the former spike, the synaptic weights increase from 11.17% to 97.23% with increasing the pulse number from 1 to 100, demonstrating versatile handing in biosimilar visual reinforcement learning. This enhancement is probably caused by superposition of mobile ions during multiple pulses with a short interval (0.5 s), which is similarly understood by the plasticity potentiation of photonic synapses made from traditional heterostructures[26,27]. After removing the light stimulus, the enhanced signals decay exponentially over time, whereas the higher peak intensity with more consecutive pulses leads to a slower decay.

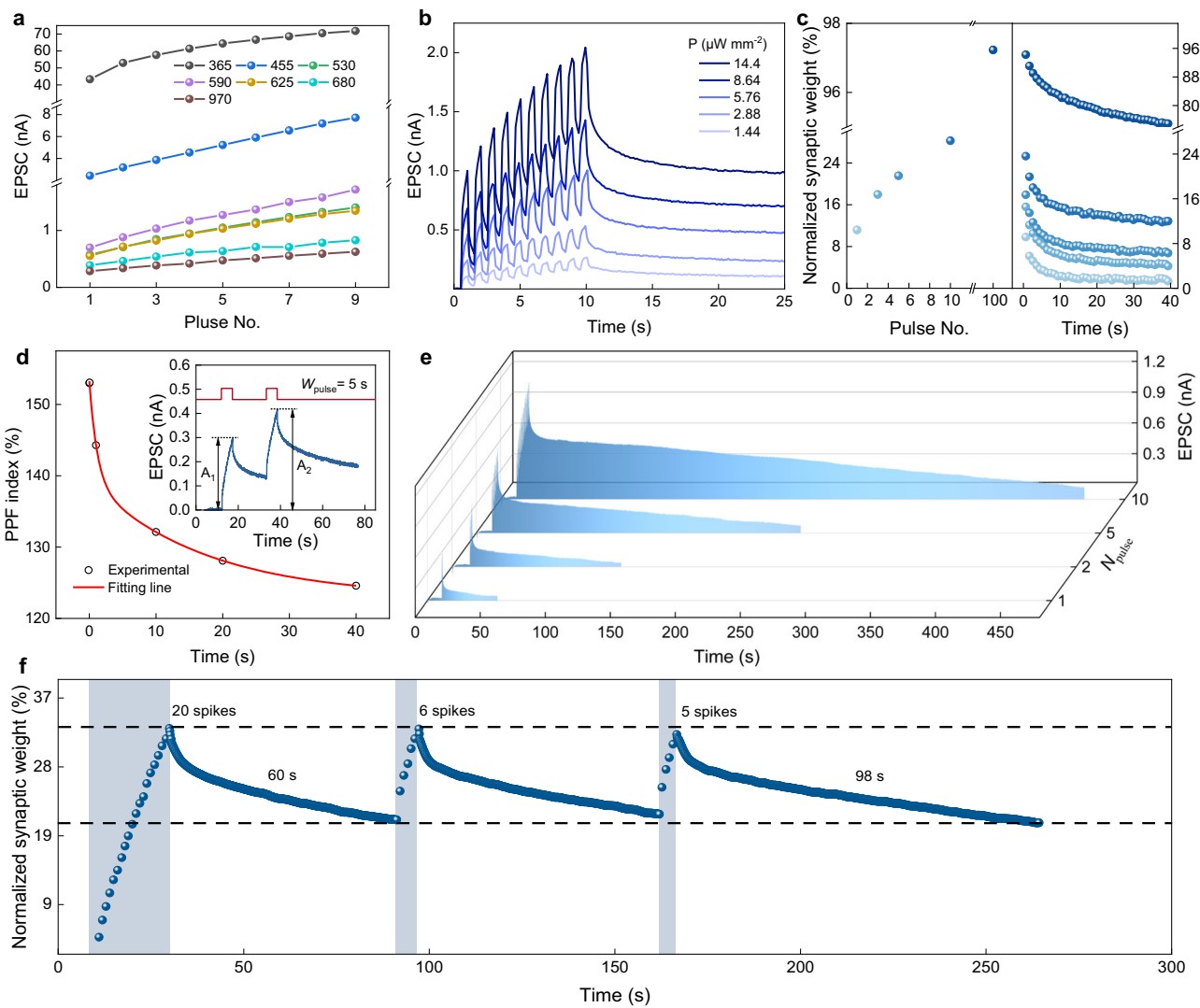

**Fig. 3 | Phototunable synaptic behaviors for retinal photoreceptor.** The essential EPSCs in response to successive light pulses. **a** Under UV-vis-NIR light sources (EPSC excitatory postsynaptic current). **b** Under light density modulation with a fixed wavelength at 365 nm and a frequency of 1 Hz (P: power density). **c** The normalized synaptic weights modulated by pulse number with a fixed power of 7.2 μW mm$^{-2}$. **d** PPF index plotted as a function of interspike interval (PPF: paired-pulse facilitation, $W_{pulse}$: pulse-width). **e** The STM to LTM transition triggered by increasing pulse number ranging from 1, 2, 5, 10 pulses ($N_{pulse}$: pulse number). **f** The measured learning-experience behaviors conducted by step-wisely multiple learning.

To quantitatively evaluate short-term synaptic plasticity, PPF index is measured by spike-time-interval-dependent plasticity (STIDP) characterization. As shown in Fig. 3d, by regulating the time intervals ranging from 0.01 to 40.00 s, the index of $A_2/A_1$ exponentially reduces with the growing interval, where $A_1$, $A_2$ are the amplitude of the first and second spike current (7.2 μW mm$^{-2}$, 365 nm). This attenuation curve can be well fitted by the double exponential function (PPF index $= C_0 + C_1 e^{-\Delta t/\tau_1} + C_2 e^{-\Delta t/\tau_2}$), that is widely followed by the bio-synapses[28]. Our zero-powered photosynapse shows a maximum PPF index of 153% for 0.01 s interval, which is comparable with those of source-driven heterostructure devices[29,30], indicating a high post-synaptic response with excitatory plasticity. Such changes in the analog electric signal can strengthen synaptic connections between neurons, leading to consolidation process of converting short-term memory (STM) into long-term memory (LTM) for efficient visual memory[31]. As described in Fig. 3e and Supplementary Fig. 10a, a temporary storage of optic information, referred as STM, lasts for a few seconds to minutes in the initial process. By extending the pulse number from 1 to 10 or increasing light $W_{pulse}$, the encoded signal can retain over an extended period of time, which eases the transition from STM to LTM.

Moreover, a stepwise learning-forgetting-relearning experience is conducted to assess the significance of successful learning (Fig. 3f and Supplementary Figs. 10b, c). With the help of 20 light pulses as the first learning stage, the photosynapse takes 20 s to approach a synaptic weight (33.5%) level that decline gradually (forgetting process) after removing the simulation. For a successive relearning on recall, it only takes 6 s to reach the previous cognitive level, meaning that revisiting cognition requires shorter time than the original learning. This trend is also observed in the third relearning stage. Notably, after the third relearning stage, the memorized information could be retained for longer periods of time and becomes more durable in comparison with the initial stage, suggesting its enhanced retention capability by repetitive reviewing stimulus. Such intriguing features are highly consistent with Ebbinghaus' theory of learning and relearning in the biology[32], making this ionogel photosynapse useful as neuromorphic photoreceptor in biomimetic retina.

## Transplantable and conformal behaviors
When constructing a desirable retina, it is also important to explore its replant and restoration function, because it can help restore vision by

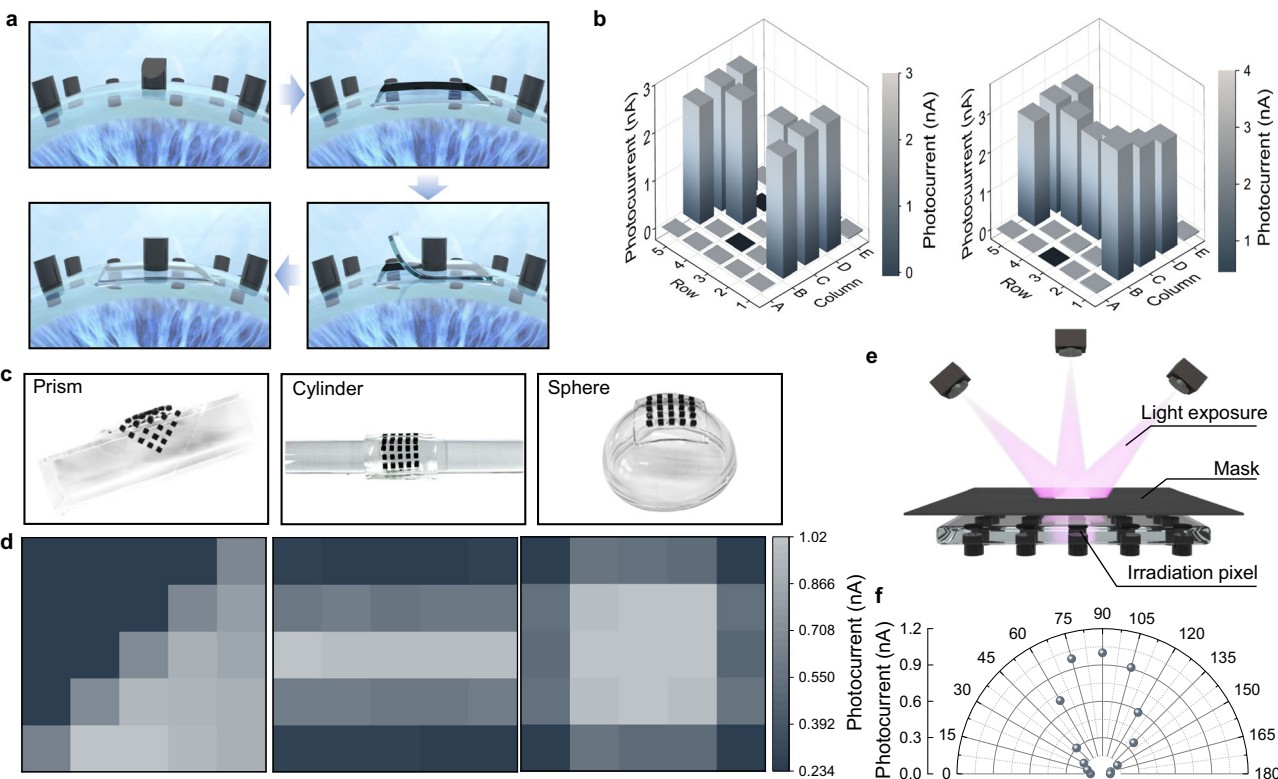

**Fig. 4 | Transplantable retinal photoreceptor with visual restoration and surface conformal. a** Flow chart of photoreceptor transplantation by replacing a damaged pix with a healthy one into host retina. **b** Visual restoration illustrated by photoelectric performance recovery from "I" mapping. **c, d** High conformal 5 × 5 pixel arrays on different surfaces and their corresponding imaging distributions under vertical light irradiation. **e** Schematic of individual photoreceptor with semi-omnidirectional light perception. **f** Angel-dependent photocurrent profile.

resolving retinal disorder or detachment and macular degeneration. Biological tissues are normally transplantable owing to their soft, regenerative and repairable properties. Inspired by those features, ionogel matters possess the dynamic and reversible multiple non-covalent interactions, including hydrogen-bonding and ionic interactions within polymer chains, exhibiting properties like high ductility (Supplementary Fig. 11), ionic transmission, and self-healing[33], which fulfill the versatile desires for emerging bioelectronics. To primarily investigate the transplantation ability of our retina, an ultrathin 5 × 5 array consisting of 25 synapse cells is developed as an artificial photoreceptor tissue. By replacing a dysfunctional or damaged native photosynapse with healthy one, the signal of photoresponse current is obviously regenerated in the repaired site under light stimuli (Fig. 4a), with a value of 2.57 nA, which is comparable with that (2.68–3.34 nA) of healthy sites (Fig. 4b). The transplanted receptor shows high stability (Supplementary Fig. 12), indicating the successful repair of defected cell. Thus, our ionogel photoreceptor holds great promise at retinal transplant and visual restoration.

With the exception of vision restoration, the biomimetic retina also has potential applications in robotic and artificial intelligence, a conformal and wide FOV capability should be needed to fuse different scenarios with complex shapes. Our retina employs the lucent ionogel as a conformal conducting membrane to implant photoreceptors and then deliver them onto the surface of arbitrary substrate, such as prism, cylinder and sphere. These membranes are highly conformal to those curvy surfaces (Fig. 4c), owing to intrinsic soft nature of gel and its strong adhesion with targets[34,35]. Figure 4d depicts their corresponding photocurrent distribution of all 25 pixels under the vertical direction homogenous exposure. The current distribution are highly depend on shape position, with a maximum values near the object corners, because these corner pixels are subject to detecting with relatively higher intensity of incident light in the three corners.

Moreover, the angle-resolved photocurrent of an isolated pixel is measured and collected with a 15 cm fixed detecting distance (Fig. 4e). By regulating incident angles between 0 and 180° with an interval of 15°, all photoresponse signals can be monitored over the whole visual field, due to the geometrical merit of bionic structure. Similar to the photoresponse of bio-eye, these FOV values are elliptically distributed (Fig. 4f), with a nonlinear increment from the horizontal to vertical direction[36].

## Neuromorphic imaging and motion tracking

As proof of this self-driven conformal retinomorphic application, we measure the real-time visual imaging of retina. Electric contacts are realized by connecting pillar array with pre-patterned copper lines on a printed hemispherical circuit board[37], a plano-convex lens (focal length, 25.9 mm) is implemented to refract and focus light on retina (Fig. 5a). The SHR-E is then interfaced with an automatic testing platform containing independent transimpedance amplifiers, computer and data acquisition card (Fig. 5b). By directly projecting an optical pattern of letter "I" onto the eye, photoreceptors covert light into neural signals, these signals are amplified into voltage signals, which are collected and then viewed through visual animation with 256-level grayscale value. Figure 5c shows the evolution of visual learning intensity of the SHR-E over pulse times. With increasing the number of training sessions (10, 20, 30 pulses), the "I" weights are continuously updated for more clarity and easier to recognize, due to the positive correlation of synaptic weight with pulse times, meaning that the image can be successfully learnt by the SHR-E through increasing learning times. Moreover, we also investigate the effect of forgetting duration on memory strength decay. After removing the illumination, the encoded image becomes blurred gradually, with a 60% level of the peak strength for 100 s decay, which can still be distinguished

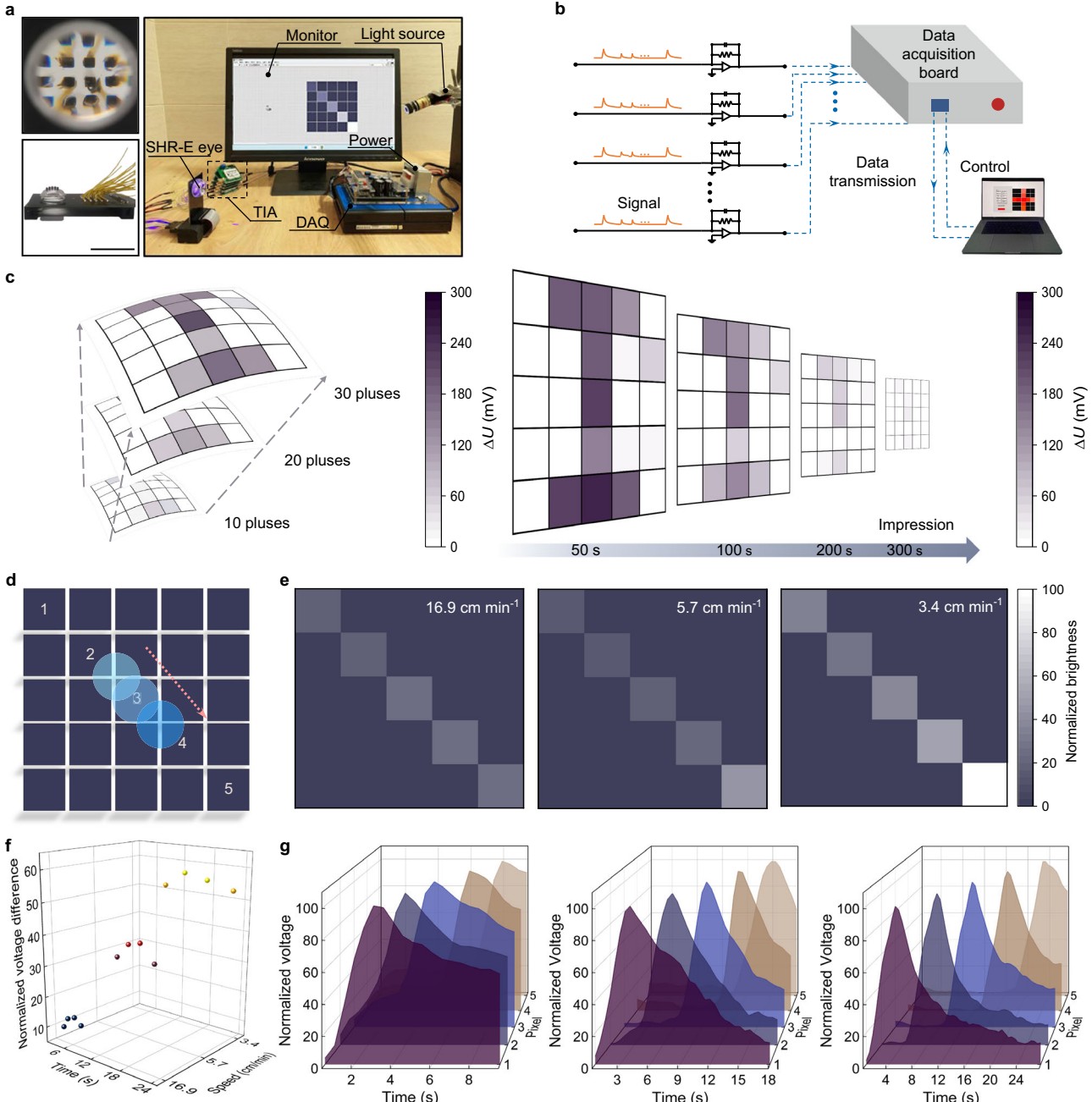

**Fig. 5 | Neuromorphic imaging and real-time motion tracking of SHR-E.**
**a** Photographs of the SHR-E mounted on circuit board and related testing platform
(SHR-E: self-powered hemispherical retinomorphic eye, TIA transimpedance
amplifier, DAQ data acquisition). Scale bar, 4 cm. **b** Schematic illustration of our
designed measurement setup. **c** Evolution of patterned images in neuromorphic
vision, including learning enhancement by increasing pulse number and memory
forgetting over the retention time ($\Delta U$: voltage variation). **d** Schematic diagram of

laser spot motion path, the arrow indicates the motion direction and the blue spot
indicates a laser with wavelength of 455 nm. **e** The real-time trajectory tracking of
motion spots with different speeds. **f** The normalized voltage difference profiles
extracted from their corresponding trajectory data. The red, green and blue color
dots correspond to the normalized voltage differences when the speeds are
16.9 cm min⁻¹, 5.7 cm min⁻¹ and 3.4 cm min⁻¹, respectively. **g** The distribution of
normalized voltage difference at anytime for speed judgment.

even after 300 s decay, demonstrating its biosimilar visual mem-
ory capability.

Furthermore, the retinomorphic vision operates by detecting
pixel-level changes in light density rather than capturing full frames
like traditional cameras. This allows for temporal information encod-
ing and processing with low-latency response, making the SHR-E sui-
table for dynamic visual imaging and motion tracking[38]. Using a 455 nm
point light source as a moving object, we collect the clustered data
from a 5 × 5 sensing array to assess its path tracking and speed mea-
surement using the automated signal acquisition system. As shown in

Fig. 5d, the overall contours of entire motion trajectory are recorded
and mapped due to the neuromorphic computing and memory of each
pixel, which tracks the desired path very well. A similar temporal
evolution of visual intensity is observed in the tracking paths with
different motion speeds. The stay points exhibit a higher brightness
than their predecessor, due to the decay of intensity over time (Fig. 5e).
This trend provides the availability of speed estimation of light spot.
Figure 5f, g shows the real-time evolution of normalized voltage
extracted from the trajectory data, where normalization processing
can eliminate the response amplitude differences of photoreceptors in

our array manufacturing. The speed is preliminarily evaluated via the difference magnitude of normalized voltage data of stay point and its synchronous value of previous point in each path, with 10.0–14.4 V for 16.9 cm min⁻¹, 29.0–36.4 V for 5.7 cm min⁻¹, and 51.7–57.7 V for 3.4 cm min⁻¹. Therefore, the neuromorphic vision enables the retina to encode sequentially temporal information for moving vehicles.

## Discussion

In conclusion, ionogel heterojunctions provide a sensing platform that is self-powered, healable, and soft. In response to light, they can act as photoreceptor to convert incident photon into neuroelectric signals with retina-like tunable plasticity. In analog to biological tissue, they can be transplantable to restore intrinsic perception function with excellent conformal capability, such as image learning and recognition as well as motion detection for biomimetic eye. Consequently, ionogel heterostructure technology offers an ideal building block for multi-stimuli responsive bionic organs in artificial intelligence.

## Methods

### Materials

Monomers of 4-Hydroxybutyl acrylate (HBA) and 2-(2-Ethoxyethoxy) ethyl acrylate (EOEOEA), the solid ionic salt bis(trifluoromethane)sulfonimide lithium salt (LiTFSI), the photoinitiator 1-Hydroxycyclohexyl phenyl ketone (HCPK), the pyrrole, and polymeric surfactant polyvinylpyrrolidone (PVP, $M_n$ = 58,000) were purchased from Sigma-Aldrich. The catalyst of ferric chloride hexahydrate ($FeCl_3 \cdot 6H_2O$) was purchased from Sinopharm Chemical Reagent Co. The covalent crosslinker ethoxylated trimethylolpropane triacrylate (ETPTA) was purchased from Curease Chemcial. All reagents were used as received without further purification.

### Synthesis of PPy nanoparticle photosensitizer

The PPy-NPs as a photosensitizer in this work were synthesized as following: 3 mmol of PVP and 14 mmol of $FeCl_3 \cdot 6H_2O$ were dissolved in a 40 mL mixed solvent of ethanol and water with a volume ratio of 1:4. Then the above solution was stirred at 1000 rpm at room temperature for 2 h. After the polymeric surfactant and catalyst were fully dissolved, a 10 mL water solution containing 6 mmol pyrrole was slowly added into the mixed solvent and stirred at 400 rpm for 4 h at room temperature. The resulting reaction solution was washed and centrifuged with ethanol for 5 times to remove any residual polymeric surfactants and catalysts, yielding a black product, i.e., PPy-NPs.

### Synthesis of ionogel and PPy nanoparticle doped ionogel

The ionogel was synthesized through a one-step co-photo-polymerization of HBA and EOEOEA monomers. In a typical procedure, the monomers HBA and EOEOEA, and cross-linker ETPTA were mixed at a volume ratio of 1:4:0.05. With the addition of photoinitiator HCPK (1% mass ratio concentration to the mixed solution) and solid ionic salt LiTFSI (0.5 M mole ratio concentration to the mixed solution), the mixed solution was stirred at 500 rpm for 30 min to obtain a uniform precursor ink, from which 2 mL of the precursor ink was poured into a polytetrafluoroethylene (PTFE) mould and exposed to 365 nm UV light at 10 W for 2 min to produce a transparent ionogel. To synthesize PPy-NPs-doped ionogel, the same procedure was followed, but with the addition of PPy-NPs to the precursor ink.

### Preparation of ionogel heterojunction

The ionogel heterojunction was prepared using a PTFE mould with a thin PTFE clapboard in the middle of mould. Firstly, the precursor ink without PPy-NPs was added into one part of the mould, and expose it to 365 nm UV light at 10 W for 2 min to obtain a transparent ionogel. Next, with removal of clapboard, another ink containing PPy-NPs was injected into the other part of mould. After irradiation with the same UV source for 4 min, a black PPy-NPs-doped ionogel was solidified, and

finally achieve the formation of a heterojunction due to the tight contact between the two ionogels.

### Fabrication of heterojunction retina

The process involved in fabricating planar-type ionogel heterojunction retina is schematically shown in Supplementary Fig. 13a. First, the ionogel precursor ink was injected into a customized PTFE mould, which had a flat groove measuring 20 × 20 mm, and was then covered with a flat plate. Next, the 1 mm thickness gel membrane was obtained by exposing it to UV light (365 nm, 10 W) for a duration of 2 min. To create the junction retina, the synthetic membrane was masked using a square hole array consisting of 5 × 5 holes, each measuring 1 × 1 mm. Subsequently, another ink doped with PPy-NPs was injected into each hole and exposed to UV light (365 nm, 10 W) for an additional 4 min. After removing the mask, a pillarforest was achieved on the top of membrane, with each pixel measuring 1 × 1 mm. Finally, copper wire electrodes were attached to both the membrane and the array, followed by welding of conductive silver paste (Ag paste) at their contact points and drying at 60 °C. Similarly, a hemispherical-type heterojunction retina was fabricated by replacing the PTFE mould with a 22 mm-diameter hemispherical polydimethylsiloxane (PDMS) mould. As shown in Supplementary Fig. 13b, a 20 mm-diameter hemispherical glass sphere was implanted into the PDMS mould with a 1 mm gap between the sphere and the mould. Then 800 μL ionogel ink was poured into the mould and irradiated under UV light (365 nm, 10 W) for 2 min to obtain hemispherical ionogel. By radially stretching the ionogel, we can transfer it onto a PET film, and leave it stuck to the PET through van der Waals interactions. Then repetition of the pillarforest fabrication process can generate the planar array after, and finally produce hemispherical retina after releasing the elastomer to relax back to its initial shape.

### Material and optoelectronic device characterization

The morphologies of PPy-NPs were characterized by field emission scanning electron microscope (JSM-7800F) at an operating voltage of 10 kV. The UV-vis-NIR absorption spectra of ionogels were recorded using a PE UV-1750 spectrophotometer. UV-vis-NIR absorption of ionogel-Ag junction was measured with a PE Lambda 950 spectrophotometer. The tensile properties were tested by a tensile-compressive tester (MDW-500 N) with a strain rate of 60 mm min⁻¹. All photoelectrical measurements were carried out through a Keithley 4200 semiconductor parameter analyzer. The light pulses with tunable wavelengths, intensities, and frequencies came from a LED driver (THORLABS, DC2200 Terminal). A series of LED sources covering wavelengths from UV, visible, and NIR (THORLABS, M365L3, M455L4, M530L4, M590L4, M680L4) were utilized. All thermal images were taken by using an infrared camera (FLIR Ti100). The optical power density is measured by an optical power meter (CEL-NP2000). A The light source system was housed in the Keithley 4200 semiconductor shielding box to prevent interference from external light signals. All electrical and optical measurements were conducted under atmospheric pressure and at room temperature.

### Visualization system

The experimental visualization setup includes a transimpedance amplifier (TIA), a Beijing Altech USB5631-D data acquisition card, a multi-circuit power supply, a laptop computer, and a Labview program. Supplementary Fig. 14 provides a schematic of the system. The power supply in this setup is utilized exclusively for operating the TIA and data acquisition card, while an external power source projects optical pattern onto the array by illuminating a mask with light. A 5 × 5 array of device is integrated with a custom printed circuit board (PCB). The data from the 25 pixels is transmitted through the PCB to the TIA for current amplification. The amplified signal is then captured by the data acquisition card and relayed to the computer for visualization.

The operation of the data acquisition card and the visualization of the data are managed using Labview software.

## Data availability
The experimental data that generated or analyzed during this study are provided with the article and its supplementary table and figures. More data are available from the corresponding author upon request.

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

## Acknowledgements
This work was supported by the National Natural Science Foundation of China (62274088, 62288102, 51703093), and the Jiangsu Specially Appointed Professor program.

## Author contributions
J.Q.L. and Z.D.L. conceived the project and designed the experiments. J.Q.L., Z.D.L. and W.H. supervised the work. X.L., C.C. and Z.H., carried out the fabrication and electrical measurements of the optoelectronic devices and SHR-E. M. W. synthesized the materials. K.P. carried out deposit and test the electrode. X.D carried out the SEM measurements. Z.F.L. and B.L. conducted the absorbance measurements and analyzed the data. C.C. and Z.Z. implemented the automatic testing platform and the visualization system. C.B.

and G.L. provided some suggestions. Y.W., R.C. and D.Z. assisted in experimental measurement. K.W., Q.W. and J.Y.L. revised the manuscript and. Z.D.L. and X.L. wrote the first draft of the manuscript, which was revised by J.Q.L. and W.H. All authors discussed the results and reviewed the manuscript.

## Competing interests

The authors declare no competing interests.
