## [Peer Review File · Nature Communications]

A bionic self-driven retinomorphic eye with ionogel photosynaptic retinaREVIEWER COMMENTS

Reviewer #1 (Remarks to the Author):

In this work by X. Luo et al., a self-powered bionic retinomorph eye is developed using selective photosensitive polypyrrole nanoparticles (PPy-NPs) doping of ionogel pillar. The results demonstrate the capability of the flexible and stretchable device of neuroelectric plasticity. The study is promising for future self-powered photo-synaptic devices. While the results are scientifically sound and of interest to the readers of Nature Comm., there are several concerns that require addressing before the paper can be considered for publication:

1- Several previous studies have explored self-powered synapses. It would be interesting to the readers to add more details about them in the introduction and a table to compare the performance and capabilities as well as the self-powering mechanism.

2- What incident power is used in fig. 2a?

3- The formula used to calculate the PPF index should be mentioned in the manuscript. Can the authors comment on the PPD index?

4- Fig. 3(a) shows the change in EPSC current with different incident light. The current for the 455 nm and 530 nm are approximately the same. What is the physical reason for getting approximately the same current?

5- Authors are advised to extend the pulsing operation to at least 100 pulses instead of 10 pulses as shown in Fig. 3(c). The low number of pulses does not provide a sufficient amount of change in the synaptic weight.

6- In Fig. 3(f), there is a clear decrement in the excitation spike from the first (20 pulses) to the second (6 pulses) excitation window. However, it is observed that there is no sufficient increment in time scale from the first (60 sec) to the second until the third excitation (which shows 98 sec). Therefore, authors need to explain the reasons behind this uncertainty in the synaptic weight changes.

7- Please elaborate on the ion migration and charge movement mechanism during potentiation and depression under different wavelengths of light to make it clearer to the reader.

8- The authors claim developing a flexible and stretchable device, however, there is no data related to the stretching capability of the device. Please show the the maximum stretching capability as well as cycling tests to confirm that the device can sustain multiple stretching cycles and thus its reliability as a physically compliant device.

9- Finally, can the authors comment on the scalability of the device as the current dimensions are quite large (1mm by 1 mm per pixel)?

Reviewer #2 (Remarks to the Author):

This manuscript introduces a self-driven hemispherical retinomorph eye (SHR-E) with an elastomeric retina, utilizing ionogel heterojunctions as photoreceptors. However, several critical aspects need more explicit clarification:

The primary claim of this research focuses on 'self-driven' photodetection. However, it is essential to note that conventional silicon photodetectors also operate at zero operational voltage, and some photodetectors with heterostructures exhibit maximum photoresponse at zero voltage.

Moreover, it would be beneficial if the authors could display the I-V curves at various light intensity levels to provide a more comprehensive understanding of the device's

performance.

The observed photoresponse appears somewhat slower compared to conventional photodetectors. It would be insightful if the authors could discuss and provide comments regarding this aspect.

Concerning the second claim related to photothermoelectric properties, there are reservations about the system's practical applicability. The skepticism stems from the notable challenges associated with temperature control in numerous scenarios.

The manuscript lacks the presentation of operational schemes (passive or active matrix operation), which impedes the visualization and understanding of the system's practical functioning.

In the introduction, there seems to be a misunderstanding concerning animal eyes. For example, stating that most biological eyes possess a hemispherical retina with wide-spectrum perception appears inaccurate. A more nuanced explanation might refer to 'photoreceptor arrays in a curved form.' Furthermore, the term 'wide spectrum perception' seems ambiguous, as the spectral range varies significantly across different species.

Response to the reviewers

Reviewer #1: *In this work by X. Luo et al., a self-powered bionic retinomorph eye is developed using selective photosensitive polypyrrole nanoparticles (PPy-NPs) doping of ionogel pillar. The results demonstrate the capability of the flexible and stretchable device of neuroelectric plasticity. The study is promising for future self-powered photo-synaptic devices. While the results are scientifically sound and of interest to the readers of Nature Comm., there are several concerns that require addressing before the paper can be considered for publication:*

Comment 1: Several previous studies have explored self-powered synapses. It would be interesting to the readers to add more details about them in the introduction and a table to compare the performance and capabilities as well as the self-powering mechanism.

Response 1: We greatly appreciate for your valuable suggestion. In recent years, self-powered photosynapses have attracted increasing attention due to their great potential in ultralow-power neuromorphic vision technologies. Up to now, several self-powered photosynapses have been constructed through different mechanisms, as shown in Table R1. It is found that the prevailing self-powered synaptic mechanisms predominantly exploit photovoltaic effects. In comparison, from the perspective of mechanism, our approach is marked by the pioneering utilization of the photothermoelectric principle, while previous self-powered photosynapses are mostly realized by photovoltaic effect. From the perspective of performances, our heterojunction ionogel not only manifests an exceptionally broad photoresponse range but also demonstrates outstanding performance with flexible and self-healable properties. Moreover, to the best of our knowledge, ionogel heterostructure in our work is reported firstly for neuromorphic vision. The Table R1 has been added as Table 1 in the revised supporting information and the comparison has been described in the revised manuscript.

Table R1. The comparison of self-powered optical synapses.

Functional layers	PPF	Wavelength (nm)	Flexibility	Healability	Self-powered mechanism	Ref.
All-inorganic CsPbI ₃ nanowire	/	405-650	/	/	Photovoltaic	Nat. Commun. 2023, 14, 1972
C ₈ -BTBT /F ₁₆ CuPc	136%	365-850	/	/	Photovoltaic	Adv. Sci. 2022, 9, 2103494
P(VDF-TrFE)/Cs ₂ AgBiBr ₆	152%	455	/	/	Photovoltaic	Adv. Sci. 2022, 9, 2106092
MAPbI ₃ :SiNCs	137%	375-808	/	/	Photovoltaic	Nano Energy 2020, 73, 104790
Spiro-OMeTAD	130%	450-650	/	/	Photovoltaic	Res. 2022, 11, 9851843
Cs ₂ AgBiBr ₆ /P(VDF-TrFE)	200%	445/660	/	/	Photovoltaic	Adv. Mater. Technol. 2023, 8, 2201779
Heterojunction ionogel	153%	365-970	√	√	Photothermoelectronic	this work

Changes in the revised manuscript:

In the introduction part, we have added the following sentences in the revised manuscript: "Recently, artificial eyes based on self-powered photosynapses have been constructed, the photoreceptors exhibit a limited light response range.....insufficient flexibility". (Page 3, Line 47-50). "which is different from the previous self-powered optical synapses induced by photovoltaic effects (**Supplementary Table S1**)," (Page 4, Line 68-69).

Comment 2: What incident power is used in fig. 2a?

Response 2: Thank you for your careful review. The optical power density used in **fig. 2a** is 7.2 $\mu\text{W}/\text{mm}^2$. The data also has been added in the revised manuscript (Page 5, Line 94).

Changes in the revised manuscript:

In the main text, we have added the incident power in the revised manuscript: "The optical power density used in **fig. 2a** is 7.2 $\mu\text{W}/\text{mm}^2$ " (Page 5, Line 94).

Comment 3: The formula used to calculate the PPF index should be mentioned in the manuscript. Can the authors comment on the PPD index?

Response 3: Thanks for your kind suggestion. The fitting formula for the PPF calculation in our work is $PPF\ index = C_0 + C_1e^{-\Delta t/\tau_1} + C_2e^{-\Delta t/\tau_2}$. Both PPF and PPD represent crucial forms of short-range plasticity in optical synapses. Based on previous literatures, the majority of two-terminal self-powered optical synapses only exhibit a single form of short synaptic plasticity due to their unidirectional responses to optical stimuli (*Nat. Commun.* 2023, 14, 1972; *Adv. Sci.* 2022, 9, 2103494; *ACS Appl. Electron. Mater.* 2023, 5, 3403), which are constant with our results. Notably, only a few studies have demonstrated the presence of both PPF and PPD capabilities in photosynapses owing to the bidirectional responses to light stimuli with different wavelengths (*Adv. Mater. Technol.*, 2023, 8, 2201779). The manifestation of these capabilities mostly rely on the optical properties of photoresponsive matter. In our study, the photosensitive materials exhibit the positive response across the entire absorption spectrum, ensuring that our device demonstrates PPF synaptic behavior without requiring an external voltage. Based on our experience and understanding, for these two-terminal self-powered photosynapse devices, it is possible to obtain the PPD function unless an external voltage is applied. However, this power loading normally results in an escalation of dark current, thereby exerting a discernible negative influence on the photoresponsibility and PPF index of the device. Therefore, how to design and fabricate the self-powered photosynapse with bidirectional modulation is highly desirable and interesting, thanks for your valuable comments.

Changes in the revised manuscript:

In the main text, we have added the fitting formula for the PPF calculation in the revised manuscript: " $PPF\ index = C_0 + C_1e^{-\Delta t/\tau_1} + C_2e^{-\Delta t/\tau_2}$ ". (Page 8, Line 154-155).

Comment 4: Fig. 3(a) shows the change in EPSC current with different incident light. The current for the 455 nm and 530 nm are approximately the same. What is the physical reason for getting approximately the same current?

Response 4: Thank you for your careful observation and suggestions. As shown in **Fig. 3a**, perhaps the similarity you are describing pertains to the light response corresponding to wavelengths of 530 nm and 625 nm. We propose that the photoelectric response of this device can be attributed to the synergistic effect between thermionic electron emission from the silver paste electrode and photothermal effect of PPy-NPs. It is well known that the LSPR of silver nanoparticles can improve the photon absorption capacity for photoresponse enhancement of photodetector. Here the silver paste's nanoparticles exhibit exceptional UV-Vis light absorption and generate a broadband photoelectric response through synergistic action. The photon energy absorbed by silver nanoparticles can release numerous electrons into the electrolyte, creating an electron-rich region on ionogel side that attracts Li^+ migration (asymmetric light absorption). At 530 nm, thermionic electron emission dominates due to the low light absorption and higher photon energy (**Figure R1a**), while at a wavelength of 625 nm, it is primarily driven by the photothermal effect induced by PPy-NPs induced temperature gradients. While the absorption of PPy-gel in Fig. R1b exhibits higher absorption at 625 nm, indicating comparable light responses within both wavelength ranges.

Figure R1. Optical absorption of materials and Localized surface plasmon resonance (LSPR) of silver nanoparticles. a, UV-Vis-NIR spectra of the silver paste. b, UV-Vis-NIR spectra of the PPy-NPs, ionogel and PPy-NPs-ionogel. The pure ionogel exhibits a narrow absorption

region at UV region while the PPy-NPs doped ionogel presents a broad absorption region from UV-Vis to NIR region. c, wavelength and light response (optical power = 2 mW), exhibiting negligible photocurrent beyond 590 nm, as shown in the absorption spectrum in a.

Comment 5: Authors are advised to extend the pulsing operation to at least 100 pulses instead of 10 pulses as shown in Fig. 3(c). The low number of pulses does not provide a sufficient amount of change in the synaptic weight.

Response 5: Thank you for your comment. According to your suggestion, the number of pulses has been expanded to 100. As shown in **Figure R2**, the photoreceptor demonstrates a more pronounced synaptic weight change when increasing the pulse number. By elevating both the intensity and frequency of spikes, the excitatory postsynaptic current (EPSC) signal levels corresponding to subsequent spikes surpass those of preceding spikes. The synaptic weights exhibit a substantial increase, escalating from 11.17% to 97.23% as the pulse count rises from 1 to 100. This observed phenomenon underscores the system's adaptability in biosimilar visual reinforcement learning. The augmentation in synaptic weights is likely attributed to the superposition of mobile ions during successive pulses with a short interval (0.5 s), a concept akin to the plasticity potentiation observed in photonic synapses constructed from conventional heterostructures. After removing the light stimulus, the enhanced signals exhibit an exponential decay over time. Notably, a higher peak intensity under more consecutive pulses leads to a slower decay. We have changed Fig. 3c in the revised manuscript.

Figure R2. The normalized synaptic weights modulated by pulse number with a fixed power of $7.2 \mu\text{W mm}^{-2}$.

Changes in the revised manuscript: In the main text, we have refreshed the **Figure 3c** and added the sentences in the revised manuscript: "the synaptic weights increase from 11.17% to 97.23% with increasing the pulse number from 1 to 100". (Page 7, Line 140-141).

Comment 6: In Fig. 3(f), there is a clear decrement in the excitation spike from the first (20 pulses) to the second (6 pulses) excitation window. However, it is observed that there is no sufficient increment in time scale from the first (60 sec) to the second until the third excitation (which shows 98 sec). Therefore, authors need to explain the reasons behind this uncertainty in the synaptic weight changes.

Response 6: We are grateful for the kind suggestion. The aim of **Figure 3f** is to simulate the learning-forgetting-relearning process. It is evident from the figure that, in order to attain the same synaptic weight, the number of optical pulse stimulations was reduced from 20 pulses to 6 pulses during the second learning stage, and further reduced to 5 pulses during the third learning stage. Meanwhile, the delay time taken to reach the same level of synaptic weight increased from 60 seconds in the first stage to 67 seconds in the second stage, ultimately reaching 98 seconds in the third stage. In comparison, we conducted an additional experimental study involving multiple pulse stimulations, where a fixed number of pulses was used in each

stage. As shown in **Figure R3** (also see **Supplementary Fig. 10c**), the results show an increase in post-synaptic current and an augmentation in synaptic weights with repeated optical pulses. Notably, each subsequent stimulation resulted in a progressively extended forgetting duration. These observed trends in synaptic weight changes align with our previous findings. This phenomenon bears resemblance to the well-known Ebbinghaus forgetting curve often observed in biological vision.

Figure R3. The “learning-forgetting-relearning” experience behavior of the photoreceptor under successively optical pulses with a fixed pulse number (365 nm, $14.4 \mu\text{W mm}^{-2}$).

Comment 7: Please elaborate on the ion migration and charge movement mechanism during potentiation and depression under different wavelengths of light to make it clearer to the reader.

Response 7: Thank you for your comment. The ion migration process can be described in detail as follows: In the wavelength range of 585 nm to 970 nm, the dominating factor is the photothermal effect of PPy-NPs. This effect leads to an increase in temperature on the side of the doped PPy-NPs gel due to thermal gradients (see **Figure R1b**). The temperature difference facilitates the migration of ions towards the undoped side, resulting in the rapid accumulation of positive charges due to the small size of Li^+ ions. In the wavelength range of 365 nm to 585 nm, the thermionic effect becomes prominent due to the exceptional UV light absorption of the silver nanoparticle paste used as contact electrodes. This effect causes a significant generation of thermoelectrons on the side of the undoped ionogel (see **Figure R1a and c**). As a result of

this charge accumulation, migrating Li⁺ ions with positive charges are attracted within the gel matrix for photocurrent signal.

Comment 8: The authors claim developing a flexible and stretchable device, however, there is no data related to the stretching capability of the device. Please show the the maximum stretching capability as well as cycling tests to confirm that the device can sustain multiple stretching cycles and thus its reliability as a physically compliant device.

Response 8: Thanks for reviewer's professional comment. We have conducted the tensile and cycling test of the ionogel junction device. As shown in **Figure R4a**, the ionogel heterojunction exhibits a stretching capability of approximately 180%, which effectively satisfies various application requirements. The stress-strain cycling curves almost overlap with its initial state even after undergoing 200 loadings at a stretch of 50%, indicating its reliable stability (**Figure R4b**). Meanwhile, the corresponding test data and relevant testing method have been added in the revised supplement information as **Supplementary Fig. 11**.

Figure R4. The mechanical property of heterojunction ionogel. (a) Tensile stress-strain curve for heterojunction ionogel. Inset: Photograph of original state and maximum strain state. (b) Cyclic stress-strain curves at a fixed strain of 50%.

Changes in the revised manuscript: In the main text, we have added the words in the revised manuscript: "**Supplementary Fig. 11**" (Page 9, Line 184). In the supplement information, we have added the Figure R4 as **Supplementary Fig. 11**.

Comment 9: Finally, can the authors comment on the scalability of the device as the current dimensions are quite large (1mm by 1 mm per pixel)?

Response 9: Thank you for your comment. To study the scalability of the ionogel device, we reduce device size to fabricate a $1\text{ cm} \times 1\text{ cm}$ array consisting of 10×10 devices by the same preparation method, which is schematically shown in **Supplementary Fig. 13**. In comparison with the same size array consisting of 5×5 devices in the manuscript, the integration density increases by four times. The photographs of the two as-fabricated array and their corresponding photoresponse properties are presented in **Figure R5**. After reducing the device geometrical dimensions, the downscaled device still exhibits photoresponse, the intensity of photocurrent is lower than that of the original device. However, due to the constraints inherent in our present functional materials and fabrication processes, the downscaled sample show a reduced yield. Failure to accomplish this integration results in reduced yields of salable devices, and therefore increased manufacturing costs. Thus, process integration is one of the biggest manufacturing challenges for realization of large-scale and high density production of downscaled ionogel devices. In the next step, we will develop advanced fabrication techniques into the overall process for high density device manufacture.

Figure R5. The scalability of the ionogel device. a-b, Photograph of a 10×10 device array at 1cm×1cm area and its photocurrent response under UV excitation. c-d, Photograph of a 5×5 device array at 1cm×1cm area and its photocurrent response under UV excitation.

Reviewer #2: *This manuscript introduces a self-driven hemispherical retinomorphing eye (SHR-E) with an elastomeric retina, utilizing ionogel heterojunctions as photoreceptors. However, several critical aspects need more explicit clarification:*

Comment 1: The primary claim of this research focuses on 'self-driven' photodetection. However, it is essential to note that conventional silicon photodetectors also operate at zero operational voltage, and some photodetectors with heterostructures exhibit maximum photoresponse at zero voltage.

Response 1: Thanks for your insightful comment. The previous studies have demonstrated that the conventional silicon or silicon heterostructures-based photodetectors exhibit exceptional photoresponse even at zero voltage (*Nature Photonics*, 2020, 7, 578–584; *Small*, 2021, 17, 2100439). In comparison with those silicon-based photodetectors, our heterojunction ionogel photoreceptor exhibits more unique properties, such as high flexibility, self-healing capabilities and excellent conformalability. Meanwhile, the ionogel photoreceptor could be fabricated by solution process method. These characteristics offer significant potential applications in various scenarios, such as humanoid robots and flexible electronics.

Comment 2: It would be beneficial if the authors could display the I-V curves at various light intensity levels to provide a more comprehensive understanding of the device's performance.

Response 2: We greatly appreciate for your valuable suggestion. In order to characterize the current-voltage (*I-V*) behavior, we performed additional experiments involving current-voltage measurement at different light intensities. As shown in **Figure R6**, by applying an external voltage, the current values are in the microampere range, which is significantly higher than the nanoampere level observed for photoresponse current in the self-driven mode (see **Figure 2a**). Therefore, with the help of power supply, the *I-V* curves exhibited minimal variation under light exposure at different light intensities.

Figure R6. *I-V* curves of the device under 365 nm light irradiation with different intensity.

Comment 3: The observed photoresponse appears somewhat slower compared to conventional photodetectors. It would be insightful if the authors could discuss and provide comment regarding this aspect.

Response 3: Thanks for your professional comment. As an emerging functional material, ionogel exhibit distinct ion conduction mechanism compared to the electron conduction in traditional materials. The conductive property of ionogels normally relies on the migration of ion, which differs from electron migration due to the substantial disparities in volume and mass between ions and electrons (*Science* 2022, 376, 502–507; *Nat. Commun.* 2019, 10, 1171). Consequently, ionic charges exhibit much slower migration speeds than electron charges under equivalent kinetic energy conditions, thereby leading to slower photoresponse compared to conventional photodetectors. We have dicussed this aspect in Page 5, Line 96-98.

Comment 4: Concerning the second claim related to photothermoelectric properties, there are reservations about the system's practical applicability. The skepticism stems from the notable challenges associated with temperature control in numerous scenarios.

Response 4: We appreciate your comment. To explore the influence of temperature on device performance. Our study involved comprehensive testing of the device's tolerance and response characteristics across different temperature ranges. Specifically, we conducted additional experiments at environmental temperatures of 20 °C, 30 °C, and 40 °C. As shown in Figure R7, temperature-dependent photocurrent behaviors were observed. With the temperature increase, their corresponding EPSC value also enhance. The observed phenomenon can be attributed to the enhanced activity of ions within the gel at the elevated temperature, thereby leading to an accelerated migration rate.

Figure R7. Effect of temperature on the photoresponse properties. a, EPSC variation curves of successively optical pulses at different temperatures (365 nm, 14.4 $\mu\text{W mm}^{-2}$). b, Sustained light at different temperatures for increased response time (365 nm, 14.4 $\mu\text{W mm}^{-2}$).

Changes in the revised manuscript: We have added the following sentence in the revised manuscript: "Furthermore, temperature-dependent photocurrent behaviors are observed in our heterostructure (**Supplementary Fig. 8**). The EPSC is more sensitive to high temperature due to the enhanced activity and accelerated migration rate of ions within the gel at elevated temperature." (Page 6, Line 126-129).

Comment 5: The manuscript lacks the presentation of operational schemes (passive or active matrix operation), which impedes the visualization and understanding of the system's practical functioning.

Response 5: Thanks for your invaluable comment. We have made a modification and additions to Methods parts of the manuscript.

Changes in the revised manuscript: In the Method part of revised manuscript, we have made a modification to Visualization system as following: "The experimental visualization setup includes a transimpedance amplifier (TIA), a Beijing Altech USB5631-D data acquisition card, a multi-circuit power supply, a laptop computer, and a Labview program. Supplementary Figure 12 provides a schematic of the system. The power supply in this setup is utilized exclusively for operating the TIA and data acquisition card, while an external power source projects optical pattern onto the array by illuminating a mask with light. A 5x5 array of devices is integrated with a custom printed circuit board (PCB). The data from the 25 pixels is transmitted through the PCB to the TIA for current amplification. The amplified signal is then captured by the data acquisition card and relayed to the computer for visualization. The operation of the data acquisition card and the visualization of the data are managed using Labview software."

Comment 6: In the introduction, there seems to be a misunderstanding concerning animal eyes. For example, stating that most biological eyes possess a hemispherical retina with wide-spectrum perception appears inaccurate. A more nuanced explanation might refer to 'photoreceptor arrays in a curved form.' Furthermore, the term 'wide spectrum perception' seems ambiguous, as the spectral range varies significantly across different species.

Response 6: Thanks for your careful checks and suggestion. According to your suggestion, we have changed the following sentence "most biological eyes possess a hemispherical retina with wide-spectrum perception appears inaccurate" into "most biological eyes possess

photoreceptor arrays in a curved form with broadband photoperception”. The changes have been highlighted in the revised manuscript.

Changes in the revised manuscript: In the introduction part of revised manuscript, we have made a modification to the relevant discription as following: “most biological eyes possess photoreceptor arrays in a curved form with broadband photoperception”. in Page 2, Line37-38.

REVIEWER COMMENTS

Reviewer #1 (Remarks to the Author):

The authors have thoroughly and successfully answered the reviewers' comments and conducted the necessary experiments to address the provided comments. The paper can now be recommended for publication.

Reviewer #2 (Remarks to the Author):

I remain cautious about current research in self-driven photodetection. For example, the authors present IV curves under varying lighting conditions, but these curves exhibit linear characteristics that hinder distinct signal differentiation. Notably, minor voltage fluctuations can cause significant changes in photocurrent, even without an increase in light exposure.

Furthermore, the current level merges at a high voltage level, rendering it unsuitable for use as a photodetector. Recently, devices under the names 'bionic eye' or 'neuromorphic devices' have been reported in various forms. Although these devices incorporate multiple characteristics and functionalities, I believe that maintaining at least the basic properties at an appropriate level is essential for the research to hold value.

Reviewer #3 (Remarks to the Author):

The work reports the self-driven ionogel based photoreceptors for artificial retina. Compositing the polypyrrole nanoparticles (PPy-NPs) into an ionogel induces the self-driven potential due to combination of photothermal effect from PPy-NPs and thermoelectric effect from ionogel. The idea looks interesting but the mechanism of the EPSC needs to be explained in more detail. It is recommended for major revision. Here are the comments for the improvement.

1. Photothermal effects due to PPy-NPs are important in generating photothermal effect. To induce the thermoelectric effect, the spatial gradient of temperature ($\Delta T(x)$) is expected to play an important role. Therefore, the authors should explain the source of the EPSC and self-driven potential in more detail with a drawing of heterojunction structure (Fig. 1c).
2. The change in the EPSC with light exposure conditions is assumed to be related to thermoelectric effect. The EPSC is measured across the ionogel. The detailed direction of ion migration, the direction of generated potential and their effect of ionic current should be explained in detail.
3. Was the EPSC the short-circuit current? In my opinion, it is required to measure the open-circuit voltage and short-circuit current to prove that the mechanism of the device is due to thermoelectric effect. The authors should also specify the source of current and how it was measured.
4. The authors mentioned that the mechanism is different from the photovoltaic effect. The explanation should be detailed in conjunction with the mechanism explanation.
5. There are some minor errors. For example, In Fig. 1a, "Cynapse" should be changed to "Synapse".

Professor Dr Juqing LIU
Key Laboratory of Flexible Electronics (KLOFE)
Institute of Advanced Materials (IAM)
Nanjing Tech University (NanjingTech)
30 South Puzhu Road, Nanjing 211816, China
Tel: + 86 25 8358 7982, Fax: + 86 25 8358 7982
Email: iamjqliu@njtech.edu.cn

Response to the reviewers

Reviewer #1: *The authors have thoroughly and successfully answered the reviewers' comments and conducted the necessary experiments to address the provided comments. The paper can now be recommended for publication.*

Response: Thank you for your positive comment.

Reviewer #2: *I remain cautious about current research in self-driven photodetection. For example, the authors present IV curves under varying lighting conditions, but these curves exhibit linear characteristics that hinder distinct signal differentiation. Notably, minor voltage fluctuations can cause significant changes in photocurrent, even without an increase in light exposure.*

Furthermore, the current level merges at a high voltage level, rendering it unsuitable for use as a photodetector. Recently, devices under the names 'bionic eye' or 'neuromorphic devices' have been reported in various forms. Although these devices incorporate multiple characteristics and functionalities, I believe that maintaining at least the basic properties at an appropriate level is essential for the research to hold value.

Response: Thanks for your comment. A self-driven or self-powered photodetector, as a new type of photodetectors, enables photodetection without the external power, has attracted great attention recently (*Nature*, 2023, 616, 712; *Nat. Commun.*, 2023, 14, 1972; *Adv. Funct. Mater.*, 2021, 31, 2011284). Compared to conventional photodetectors, the optoelectronic synapses can detect and memorize the optical signals simultaneously, thus enabling the implementation of a biological vision and an optogenetic neural network (*Nat. Nano.*, 14, 776, 2019; *Nat. Nano.*, 17, 27, 2022). In our study, the ionogel heterojunction device exhibits self-driven optoelectronic synapse function, which can convert light signals into electric signals and process the signals with the synaptic weight without any external power requirements. With the help of light exposure, the ionogel heterojunction generates a temperature gradient between pure-gel and PPy-gel, owing to the photothermal effect. This temperature gradient can drive thermodiffusion of mobile ions, resulting in an ion concentration gradient that produces a low electric voltage in the millivolt range. Generally, the process that converting temperature difference into electric voltage is defined as the thermoelectric effect, also known as the Soret effect or ionic Seebeck effect (*Science* 2020, 368, 1091; *Sci. Adv.*, 2022, 8, eabq8432). Therefore, our heterojunction device is considered as a self-driven photosynapse instead of photodetection.

To understand the I-V behaviors of heterojunction device under an applied voltage (It is noted that we don't need the applied external voltage in our manuscript), here we conducted the current-voltage measurement experiment again under varying light intensities (**Fig. R1**), the results is very similar to that in the first revision. A plausible explanation for this phenomenon is illustrated as follows: The heterogel normally acts as an ionic conductive material, wherein a temperature gradient within the heterogel induces ion migration from hot side to cold side in absence of external voltage. This thermodiffusion process generates ionic current, with an order of nA. However, when an external large voltage is applied, ion movement driven by electric field becomes dominant, with an order of μA . Therefore, it can be concluded that the ion movement driven by photothermal effect plays a subordinate role during the applied voltage. As a result, when an external voltage is introduced, no appreciable changes in current are seen under varying light illumination.

Fig. R1. I-V curves of device under 365 nm light irradiation with different intensity at 0-5 V

Futhermore, the SNR curves reflect the disparity between signal and noise, and a higher value of this parameter enhances the selectivity of the device for optical detection (*Adv. Mater.*, 2023, 35, 2209004; *Adv. Funct. Mater.*, 2022, 32, 2207713). In contrast to conventional

semiconductor materials that employ electrons and holes as charge carriers, ion gels represent a novel conductor utilizing ions as transport carriers. The dark current of ion gels surges with increasing bias voltage. Therefore, testing the device at low voltage is necessary to achieve low dark current levels. Our device can be tested at zero bias voltage which not only conserves energy but also yields minimal dark current while enhancing optical responsiveness.

Fig. R2. Signal-to-noise ratio curves of the device at different voltages.

Reviewer #3: *The work reports the self-driven ionogel based photoreceptors for artificial retina. Compositing the polypyrrole nanoparticles (PPy-NPs) into an ionogel induces the self-driven potential due to combination of photothermal effect from PPy-NPs and thermoelectric effect from ionogel. The idea looks interesting but the mechanism of the EPSC needs to be explained in more detail. It is recommended for major revision. Here are the comments for the improvement.*

Comment 1: Photothermal effects due to PPy-NPs are important in generating photothermal effect. To induce the thermoelectric effect, the spatial gradient of temperature ($\Delta T(x)$) is expected to play an important role. Therefore, the authors should explain the source of the EPSC and self-driven potential in more detail with a drawing of heterojunction structure (Fig. 1c).

Response 1: We greatly appreciate for your valuable suggestion. We have incorporated the following modifications in Fig. 1c to enhance the comprehensibility (**Fig. R3**). The middle figure of Fig. 1c now includes annotations indicating the direction of ion migration driven by the photothermal effect and the direction of the built-in electric field. Under light illumination, ions migrate from the PPy-gel to the pure-gel within heterogel owing to the photothermal effect, resulting in an imbalanced ion concentration due to the different migration rates between cations and anions. The imbalance can produce a built-in electric field within the heterogel, directed from pure-gel towards PPy-gel. Furthermore, when light illumination is removed, the temperature gradient is eliminated. Ions will diffuse from the pure-gel side to the PPy-gel side, which is attributed to the difference in ion concentration in both sides. Consequently, we have also included illustrations depicting ion relaxation in the revised figure of Fig. 1c.

Changes in the revised manuscript: In the main text, we have refreshed the **Fig. 3c**. We have explain the mechanism in detail in Page 4, Line 85-96.

Original Fig. 1c:

Revised Fig. 1c:

Fig. R3. The original and revised Fig. 1c.

Comment 2: The change in the EPSC with light exposure conditions is assumed to be related to thermoelectric effect. The EPSC is measured across the ionogel. The detailed direction of ion migration, the direction of generated potential and their effect of ionic current should be explained in detail.

Response 2: Thanks for your kind suggestion. It is well known that both the cations and anions exhibit different mobilities. When a temperature gradient is established between pure-gel and PPy-gel under light illumination, both the cations and anions migrate from hot side (PPy-gel) to cold side (pure-gel). With the help of this thermodiffusion process of ions, ionic currents are generated. However, compared to TFSI⁻ ions, Li⁺ ions demonstrate higher mobility due to their smaller size. Consequently, a larger number of Li⁺ ions infiltrate into the pure-gel, resulting in an excess of positive ions over negative ions. Conversely, there are more negative ions than positive ions presented in PPy-gel. The impact of thermodiffusion on ion flow was investigated by measuring the ionic conductivity before and after light irradiation for each region (**Fig. R4** and Supplementary Fig. 5 in revised SI). Significant changes in conductivity were observed with

a reduction of $9.48 \times 10^{-5} \text{ S m}^{-1}$ in the doped region and a growth of $0.96 \times 10^{-5} \text{ S m}^{-1}$ in the undoped region as a result of ion drift from the hot (dopant) side towards the cool (pure) side, which certified the light-driven ionic migration induced by photothermoelectric effect. A built-in electric field is created in the heterogel, pointing from the pure-gel toward the PPy-gel, which impedes Li^+ ions diffusion while accelerating TFSI^- ions diffusion until equilibrium is reached. Ultimately, a state of equilibrium is achieved between the temperature gradient effect and the driving force exerted by the electric field.

Fig. R4. Ionic conductivity of the heterogel, pure-gel, and PPy-gel under light exposure.

Changes in the revised manuscript: In the main text, we have explain the mechanism in detail in Page 4, Line 85-96.

Comment 3: Was the EPSC the short-circuit current? In my opinion, it is required to measure the open-circuit voltage and short-circuit current to prove that the mechanism of the device is due to thermoelectric effect. The authors should also specify the source of current and how it was measured.

Response 3: Thank you for your careful review and kind suggestion. The EPSC was acquired at a 0 V bias, thus it can be regarded as the short-circuit current. As depicted in **Fig. R5a**, the device exhibits an open-circuit voltage of 35 mV and a short-circuit current of 19 nA under 365 nm light ($14.4 \mu\text{W mm}^{-2}$). A linear correlation exists between the open circuit voltage (ΔU) and

temperature (ΔT), expressed as $\Delta U = \Delta T \cdot S$. Here, S represents the ionic Seebeck coefficient of the gel which is determined to be 1.57 mV K^{-1} (**Fig. R5b**). The current was measured using the Keithley 4200A-SCS parameter analyzer under various light intensities with a bias of 0 V.

Fig. R5. a, Short circuit current and open circuit voltage of the device. b, ΔU as a function of ΔT .

Comment 4: The authors mentioned that the mechanism is different from the photovoltaic effect. The explanation should be detailed in conjunction with the mechanism explanation.

Response 4: Thank you for your kind suggestions. The working mechanism of our device is based on the Soret effect (or ionic Seebeck effect), where a temperature gradient induces an inhomogeneous distribution of cations and anions, resulting in a built-in electric field and ionic current. The fundamental operating mechanism of most self-powered photodetectors that rely on traditional semiconductors is the photovoltaic effect. This effect enables direct harvesting of solar energy by converting incident photons into a flow of free charge carriers. It naturally occurs in semiconductor junction-based devices. The photovoltaic effect typically involves two primary processes: absorption of incident photons by the photoactive material leading to excitation of electron-hole pairs, followed by separation and transfer of these pairs to the electrodes driven internally within the photoelectric device. The mechanism explanation is explained in detail in Page 6, Line 116-127.

Professor Dr Juqing LIU
Key Laboratory of Flexible Electronics (KLOFE)
Institute of Advanced Materials (IAM)
Nanjing Tech University (NanjingTech)
30 South Puzhu Road, Nanjing 211816, China
Tel: + 86 25 8358 7982, Fax: + 86 25 8358 7982
Email: iamjqliu@njtech.edu.cn

Comment 5: There are some minor errors. For example, In Fig. 1a, “Cynapse” should be changed to “Synapse”.

Response 5: Thanks for your careful checks and suggestion. According to your suggestion, we have changed the “Cynapse” to “Synapse”.

REVIEWERS' COMMENTS

Reviewer #2 (Remarks to the Author):

The revised version is well-written based on all the reviewer's concerns.

Reviewer #3 (Remarks to the Author):

Most of the comments raised by the reviewer have been addressed. The paper is recommended for publication after minor revision. In Figure 1, please enhance the schematic of the device structure by providing more clarity on the electrodes used for measurements.

Professor Dr Juqing LIU
Key Laboratory of Flexible Electronics (KLOFE)
Institute of Advanced Materials (IAM)
Nanjing Tech University (NanjingTech)
30 South Puzhu Road, Nanjing 211816, China
Tel: + 86 25 8358 7982, Fax: + 86 25 8358 7982
Email: iamjqliu@njtech.edu.cn

Response to the reviewers

Reviewer #2: *The revised version is well-written based on all the reviewer's concerns.*

Response: Thank you for your positive comment.

Reviewer #3: *Most of the comments raised by the reviewer have been addressed. The paper is recommended for publication after minor revision. In Figure 1, please enhance the schematic of the device structure by providing more clarity on the electrodes used for measurements.*

Response: We greatly appreciate for your valuable suggestion. During the device preparation and testing process, we utilized silver paste (Ag paste) as the conductive and connecting layer. Consequently, in Figure 1a of the revised manuscript, two Ag electrodes were incorporated on both sides of the pure and doped ionogel. The modified figure is presented below.